# Identification of trypanosomatids and blood feeding preferences of phlebotomine sand fly species common in Sicily, Southern Italy

Jessica Maria Abbate[1], Carla Maia[2], André Pereira[2], Francesca Arfuso[1], Gabriella Gaglio[1], Maria Rizzo[1], Giulia Caracappa[1], Gabriele Marino[1], Matthias Pollmeier[3], Salvatore Giannetto[1], Emanuele Brianti[1]*

1 Department of Veterinary Sciences, University of Messina, Messina, Italy, 2 Global Health and Tropical Medicine (GHTM), Institute of Hygiene and Tropical Medicine (IHMT), New University of Lisbon, Lisbon, Portugal, 3 Bayer Animal Health GmbH, Leverkusen, Germany

* ebrianti@unime.it

**Data Availability Statement:** Obtained sequences were deposited at the DNA Data Bank of Japan (DDBJ) (http://www.ddbj.nig.ac.jp/), and the following Accession numbers were assigned:

## Abstract

In this study, the presence of *Leishmania* DNA and blood feeding sources in phlebotomine sand fly species commonly present in Sicily were investigated. A total of 1,866 female sand flies including 176 blood fed specimens were sampled over two seasons in five selected sites in Sicily (southern Italy). *Sergentomyia minuta* (*n* = 1,264) and *Phlebotomus perniciosus* (*n* = 594) were the most abundant species at all the sites, while three other species from the genus *Phlebotomus* (*i.e.*, *P. sergenti n* = 4, *P. perfiliewi n* = 3 and *P. neglectus n* = 1) were only sporadically captured. Twenty-eight out of the 1,866 (1.5%) sand flies tested positive for *Leishmania* spp. *Leishmania tarentolae* DNA was identified in 26 specimens of *S. minuta*, while the DNA of *Leishmania donovani* complex was detected in a single specimen each of *S. minuta* and *P. perniciosus*. Interestingly, seven *S. minuta* specimens (0.4%) tested positive for reptilian *Trypanosoma* sp. Blood sources were successfully identified in 108 out of 176 blood fed females. Twenty-seven out of 82 blood sources identified in fed females of *P. perniciosus* were represented by blood of wild rabbit, *S. minuta* mainly fed on humans (16/25), while the sole *P. sergenti* fed specimen took a blood meal on rat. Other vertebrate hosts including horse, goat, pig, dog, chicken, cow, cat and donkey were recognized as blood sources for *P. perniciosus* and *S. minuta*, and, surprisingly, no reptilian blood was identified in blood-fed *S. minuta* specimens. Results of this study agree with the well-known role of *P. perniciosus* as vector of *L. infantum* in the western Mediterranean; also, vector feeding preferences herein described support the hypothesis on the involvement of lagomorphs as sylvatic reservoirs of *Leishmania*. The detection of *L. donovani* complex in *S. minuta*, together with the anthropophilic feeding-behaviour herein observed, warrants further research to clarify the capacity of this species in the transmission of pathogens to humans and other animals.

LC464942-LC464968; LC469805-LC469912; LC471393-LC471407 (http://getentry.ddbj.nig.ac.jp).

**Funding:** Sand fly collection in 2018 season has been partially funded by Bayer Animal Health. The founder did not play any role in the study design, data collection and analysis, decision to publish and preparation of the manuscript. Molecular analysis on sand flies were partially founded by the grant Research and Mobility no.015063 awarded by the University of Messina. Dr Carla Maia has been awarded with Investigator Starting Grant F/01302/2015, and André Pereira received PhD grant SFRH/BD/116516/2016. Bayer Animal Health provided support in the form of salaries for authors [MP], but did not have any additional role in the study design, data collection and analysis, decision to publish, or preparation of the manuscript."

**Competing interests:** The authors have declared that no competing interests exist. The commercial affiliation of MP does not alter our adherence to PLOS ONE policies on sharing data and materials.

# Introduction

Phlebotomine sand flies (Diptera: Phlebotominae) are insects of great interest in human and veterinary medicine. They are vectors of viral and bacterial pathogens and recognized as the main hematophagous arthropods proven to transmit protozoa of the genus *Leishmania* such as *Leishmania infantum*, the causative agent of canine leishmaniosis (CanL) in dogs and visceral (VL) or cutaneous (CL) leishmaniosis in humans in the Mediterranean area [1].

Sand flies are distributed throughout many regions of the world and their biodiversity and phenology have been investigated particularly in *Leishmania* endemic areas. In western Europe, only sand flies of the genus *Phlebotomus* are competent vectors for *Leishmania* transmission and *P. perniciosus* is the most widespread species [2].

Surveillance on phlebotomine sand fly vectors is pivotal to assess the risk for transmission of endemic *Leishmania* species, but also crucial to monitor the risk for introduction of new *Leishmania* species in non-endemic territories [2–3].

Several studies have investigated the presence of *Leishmania* DNA in phlebotomine sand flies, and the reported infection rate in *P. perniciosus* varied from 0.13% to 50% depending on the epidemiological context [4–5]. Although *L. infantum* is the only species widespread in Europe that causes illness, human cases due to non-indigenous *Leishmania* species are increasingly reported [6].

In Italy, imported leishmaniosis cases are mainly associated with international travellers and/or refugees coming from endemic zones, and consist of chronic forms diagnosed many months after entering the country [7–8]. Noteworthy, anthroponotic species such as *Leishmania tropica* and *Leishmania donovani* are included among the causative agents of imported human cases, and the risk of introduction and/or wider establishment of these species into Europe is higher since suitable sand fly vectors are present [3,6,9].

Sicily, one of the two major Italian islands, is located in the centre of the Mediterranean Sea. The island is characterized by a typical temperate climate, with mild and wet winters and hot dry summers. It is a well-known hyper-endemic region for CanL with a prevalence of infection up to 40% in dogs which are regarded as the main domestic reservoir hosts and source for human infection [10–11]. According to available entomological surveys on *L. infantum* vectors in Sicily, *P. perniciosus* has been recognized as the most abundant species, with *P. neglectus* and *P. perfiliewi* much less frequently detected [12–14]. The presence of *P. sergenti* has also been sporadically reported along the east side of the Island [12–14] implying the risk for *L. tropica* transmission. Although no autochthonous cutaneous leishmaniosis cases (ACL) caused by *L. tropica* have been detected in south-western Europe so far, Sicily is an area potentially susceptible to the introduction of this protozoan species due to its proximity to northern Morocco, an emerging ACL area [15], and to the large migratory flow of people between the two territories. As matter of fact, the flow of immigrants from endemic regions of North Africa and the Middle East, together with the rising number of reports of *L. tropica* human infections in Italy [6,7,16–17], could constitute the first step in its potential spread in Sicily and subsequently across other areas of southern Europe where the competent vector has been described [7].

Female sand flies are hematophagous insects and take blood meals on many vertebrate hosts as reptiles, birds, a variety of domestic and wild mammals and humans, as well showing an opportunistic feeding behaviour [1,4,18–19]. Interestingly, lagomorphs have been found to be the most frequent blood source for *P. perniciosus* in a large outbreak of human leishmaniosis in Madrid [20–21], and, xenodiagnostic studies demonstrated the role of hares and wild rabbits as reservoir hosts of *L. infantum*. Together these findings suggest the possible involvement of lagomorph species in the epidemiology of *Leishmania* infection [22–25]. Therefore, the identification of blood meal sources in wild-caught sand flies provides information on

host-feeding patterns under natural conditions, which, in turn, results in data on potential reservoir hosts and essential knowledge for the establishment of efficient control strategies.

Given all of the above considerations, the aim of the present study was to investigate *Leishmania* infection rates and blood sources of phlebotomine sand fly species caught in Sicily over two transmission seasons.

## Materials and methods

### Sand fly collection and identification

Sand flies in this study were sampled during two different entomologic surveys carried out in privately owned areas located in Sicily in 2017 and 2018 (Fig 1). In 2017, sand flies were captured in a suburban area (Site A. 38°13'59" N; 15°32'49" E; 263 m.a.s.l) nearby the didactic farm of the Department of Veterinary Science of the University of Messina using a classical light trap (CLT), equipped with a traditional incandescent lamp (12V, 8W) and five Laika 4.0 light traps with LED of different colours (Fig 1). Traps were placed from before sunset until

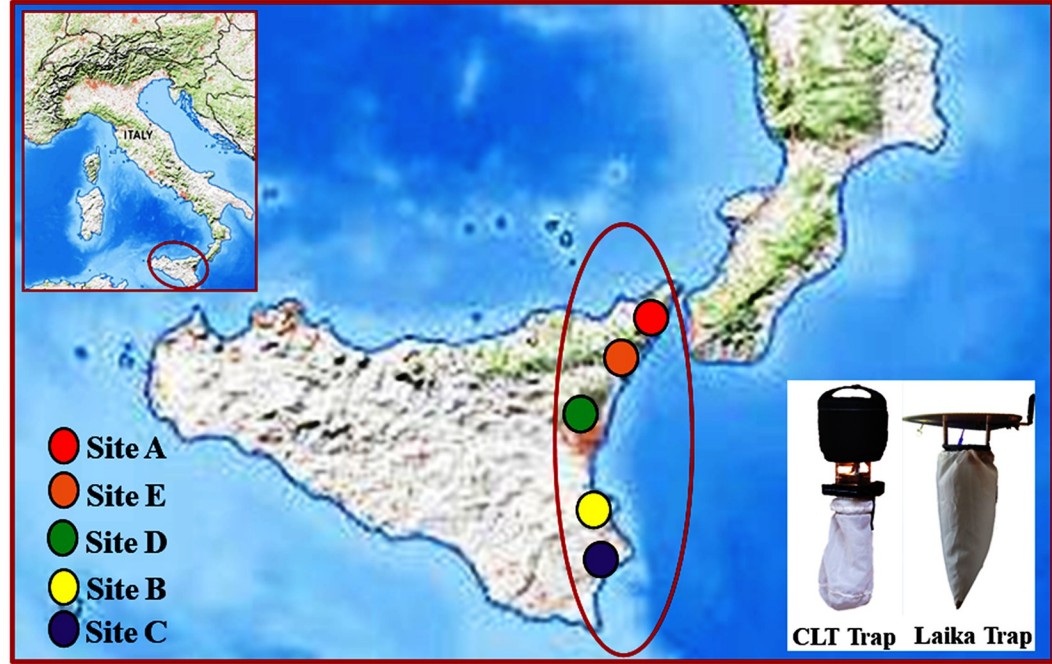

| Sand fly collection sites | Traps | Collection season | Environment | Vertebrate hosts presence in 100 meters radius |
|---|---|---|---|---|
| Site A | Laika/CLT | 2017 | Suburban | Dogs, horses, humans |
| Site B | CLT | 2018 | Suburban | Cats, dogs, humans, pigs |
| Site C | CLT | 2018 | Suburban | Cats, dogs, humans |
| Site D | CLT | 2018 | Suburban | Dogs, humans |
| Site E | CLT | 2018 | Rural | Cats, cows, dogs, goats, horses, humans, rabbits, chickens, pigs |

**Fig 1. Geographical characteristics of sand fly sampling sites and presence of vertebrate hosts.**

sunrise for three consecutive nights each month from May to October [26]. In 2018, sand fly captures were performed in four different sites, *i.e.*, in three shelters located in suburban areas of Syracuse (Site B. 37˚04'54" N; 15˚12'43" E; 69 m.a.s.l) (Site C. 37˚04'30" N; 15˚13'59" E; 416 m.a.s.l) and Catania (Site D. 37˚36'45" N; 15˚01'46" E; 948 m.a.s.l) and on a farm situated in a rural area of the municipality of Messina (Site E. 38˚00'42" N; 15˚25'18" E; 177 m.a.s.l). In each of the above sites, light traps (*i.e.*, CLT) were placed from May to October and left working from before sunset until sunrise for two consecutive nights twice a month (Fig 1).

Access to the collection sites and sampling procedures were authorized by owners of the farms (Site A and Site E) and of the shelters (Sites B, C and D).

In the laboratory, sand flies were differentiated by sex with the aid of a stereomicroscope and thereafter processed for identification. Briefly, the head and posterior last tergites of females were dissected, cleared and slide-mounted for microscopic observation as described elsewhere [12], and identified to species level using morphological keys [27]. Females were further ranked as unfed (no visible blood in the abdomen), blood fed (presence of blood in the abdomen) and gravid (presence of eggs). Finally, the thorax and the abdomen of each specimen were individually transferred into 2 mL vials containing 70% ethanol and stored for DNA analysis.

## DNA extraction and *Leishmania* spp. detection

Genomic DNA was extracted from portions (*i.e.*, thorax and abdomen) of each sand fly specimen stored in 70% ethanol using the Citogene® Cell and Tissue kit (Citomed, Portugal) according to the manufacturer's instructions. The DNA extracted was suspended in 30 μL of sterile water and stored at + 4˚C until analysis.

**Trypanosomatidae DNA amplification.** The presence of Trypanosomatidae DNA in unfed, blood fed and gravid females was firstly screened using a one-step PCR protocol with a set of primers targeting sections of the ribosomal internal transcribed spacer (*ITS-rDNA*) [28]. For further molecular characterisation of *ITS*-rDNA positive samples, a two-step PCR protocol using a set of primers targeting conserved regions of the mitochondrial cytochrome b gene (*cytB*) was performed [29]. An additional one-step PCR with specific primers for small subunit ribosomal DNA gene (*SSU*-rDNA) partial amplification [30] was carried out, to characterise the samples where the presence of *Trypanosoma* spp. was suggested by BLAST analysis of *cytB* sequences (S1 Table).

## Blood source identification

Identification of blood sources was conducted by the amplification of a 350 bp segment of the host mitochondrial *cytB*, using the modified vertebrate-universal specific primers (cytB1-F and cytB-2-R) on blood fed sand fly specimens [31]. The *cytB* PCR was carried out with 5 μL of extracted DNA in a final volume of 25 μL, using 12.5 μL of NZYTaq 2× Green Master Mix (Nyztech, Portugal) and 1.5 μL of each primer (10 pmol/μL). Amplification was performed as follows: one cycle at 94˚C for 5 min, followed by 40 cycles consisting of denaturation at 94˚C for 1 min, annealing at 55˚C for 1 min and elongation at 72˚C for 1 min, followed by final elongation at 72˚C for 7 min [4] (S1 Table). Electrophoresis of PCR products was carried out in 1.5% agarose gel stained with 2.5 μL Greensafe premium® (Nzytech, Portugal), using a 100 bp DNA ladder as a molecular weight marker and final amplicons were visualized under UV light.

## Sequence analysis

PCR products were purified and sequenced by Sanger's method (StabVida, Portugal), using the same primers used for the PCR reactions. Nucleotide sequences obtained were examined using 4Peaks v1.8 (Nucleobytes, Netherlands) and analysed by BLAST (http://blast.ncbi.nlm.

nih.gov/Blast.cgi) to find positional homologs. Obtained sequences were deposited at the DNA Data Bank of Japan (DDBJ) (http://www.ddbj.nig.ac.jp/).

### Phylogenetic analysis

Multiple sequence alignments of nucleotide datasets were performed using the iterative G-INS-I method as implemented in MAFFT v7 [32]. The obtained alignments were optimized via Gblocks [33], followed by their manual correction considering the encoding reading frame. Phylogenetic trees were constructed using the Maximum Likelihood method under the best-fitting evolutionary model (GTR + G +I; GTR—general time reversal, G—gamma distribution, I —proportion of invariant sites), selected based on the corrected Akaike information criterion, as suggested by Mega v6 [34]. The stability of the obtained tree topologies was assessed by bootstrapping with 1000 resamplings of the original sequence data. The generated trees were edited for display using FigTree v1.4.3 (available at http://tree.bio.ed.ac.uk/software/figtree/).

### Statistical analysis

Statistical analysis was performed only for data from sites where a large number of sand flies was captured. Variables were tested for normality of distribution using the Shapiro-Wilk test. As assumption of normality was not valid ($P < 0.05$), nonparametric analysis was carried out. Chi-square analysis was performed to evaluate the difference in host preference within and between sand fly species. A $P$ value $< 0.05$ was considered as statistically significant. Data were analyzed using statistical software Prism v4.00 (GraphPad Software Ltd., USA).

## Results

### Sand fly identification and *Leishmania/Trypanosoma* infection rates

During the two entomological surveys, a total of 3,090 sand flies ($n = 1,866$ females and $n = 1,224$ males) were captured. Species and details on phlebotomine sand flies collected in each site are summarized in Table 1. Among the female sand flies, *S. minuta* was the most abundant species ($n = 1,264$; 67.7%); followed by four species from the genus *Phlebotomus*: *P. perniciosus* ($n = 594$; 31.8%); *P. sergenti* ($n = 4$; 0.2%); *P. perfiliewi* ($n = 3$; 0.2%) and *P. neglectus* ($n = 1$; 0.1%). One hundred seventy-six females were blood fed (9.4%) and 113 were gravid (6.1%) (Table 2).

   *Leishmania* DNA was detected in twenty-eight female sand flies out of 1,866 (1.5%). In detail, 26 out of 1,264 (2.1%) *S. minuta* specimens ($n = 21$ unfed; $n = 4$ blood-fed and $n = 1$ gravid) tested positive to *Leishmania tarentolae* DNA; whereas *Leishmania donovani* complex DNA was identified in a *S. minuta* unfed female (1/1264; 0.1%) and in a *P. perniciosus* blood-

**Table 1. Total sand flies collected during the two entomological surveys.**

| Site (Year) | Sand fly species | | | | | | | | | |
| --- | --- | --- | --- | --- | --- | --- | --- | --- | --- | --- |
| | *Sergentomyia minuta* | | *Phlebotomus perniciosus* | | *Phlebotomus sergenti* | | *Phlebotomus perfiliewi* | | *Phlebotomus neglectus* | |
| | **Females** | **Males** | **Females** | **Males** | **Females** | **Males** | **Females** | **Males** | **Females** | **Males** |
| Site A (2017) | 62 | 48 | 157 | 141 | | | | | 1 | 2 |
| Site B (2018) | 154 | 99 | 54 | 81 | | | | | | 2 |
| Site C (2018) | 153 | 172 | 1 | 22 | 1 | | | | | |
| Site D (2018) | 248 | 162 | 9 | 78 | 3 | 5 | | | | 2 |
| Site E (2018) | 647 | 144 | 373 | 239 | | | 3 | | | 27 |
| **Total** | 1,264 | 625 | 594 | 561 | 4 | 5 | 3 | | 1 | 33 |

fed female (1/594; 0.2%). Seventeen out of the 26 *L. tarentolae* positive females (*n* = 13 unfed; *n* = 3 blood-fed and *n* = 1 gravid) were captured in the rural biotype (Site E), whereas 9 (*n* = 8 unfed and *n* = 1 blood-fed) were collected in periurban environments (Sites A-D). Both *L. donovani* complex positive females (*n* = 1 unfed and *n* = 1 blood-fed) were captured in the rural environment (Site E).

The 26 *cytB* sequences obtained from *S. minuta* revealed >99% sequence identity and 100% sequence coverage with reference sequences of *L. tarentolae* (accession number: LC092878). The *cytB* sequences obtained from *S. minuta* and *P. perniciosus* specimens, revealed > 99% identity and 100% sequence coverage with *L. donovani* complex sequences (accession numbers: CP022652; KX061917). All *ctyB*-rDNA obtained sequences were submitted to DDBJ (DDBJ Accession numbers: LC464942 to LC464968; LC471400). Based on phylogenetic analysis, 24 out of 26 obtained *L. tarentolae* sequences amplified from *S. minuta*, segregate together with *L. tarentolae* reference sequences in a monophyletic cluster, supported by a high bootstrap value (*i.e.*, 99). Despite the heterogeneity of the sequences, phylogenetic analysis has not evidenced an unambiguous existence of different haplotypes. The two *L. donovani* complex sequences herein amplified from *S. minuta* and *P. perniciosus* extracts, segregate in the *L. donovani* complex cluster (Fig 2). Based on cytB phylogenetic tree, the sequences obtained from both specimens segregate independently. The existence of intra-groups was suggested within the *L. donovani* complex cluster, however none of them evidenced the monophyly of *L. donovani/L. infanutm* species.

In addition to *Leishmania*, *Trypanosoma* DNA was detected in seven specimens of *S. minuta*, captured in rural Site E (*n* = 1) and periurban biotopes (Site A, *n* = 3; Site B, *n* = 1; Site C, *n* = 2). The 7 *cytB* sequences obtained revealed > 95% sequence identity and a sequence coverage > 80% with a reference sequence of *Trypanosoma lewisi* of Chinese origin (accession number: KR072974) [35]. The sequences were submitted to DDBJ (DDBJ Accession numbers: LC471393 to LC471399). The *SSU*-rDNA nucleotide sequences showed both 100% identity and coverage with the sequence of *Trypanosoma* sp. isolated from Gecko *Tarentola annularis* (accession number: AJ620548) [36]. The *SSU*-rDNA obtained sequences were submitted to DDBJ (DDBJ Accession numbers: LC471401 to LC471407). The phylogenetic tree shows that the obtained sequences share the same common ancestry of *Trypanosoma varani* (Accession number: AJ005279) forming together with *Trypanosoma* sp. isolated from Gecko a monophyletic cluster supported by a high bootstrap value (*i.e.*, 92); whereas two sequences are different. These two sequences also share the same common ancestry of *T. varani* but segregate independently (Fig 3).

**Table 2. Female sand flies sampled and molecularly analysed.**

| Site (Year) | Female sand fly species | | | | | | | | | | | | | | |
|---|---|---|---|---|---|---|---|---|---|---|---|---|---|---|---|
| | *Sergentomyia minuta* (*n* = 1,264) | | | *Phlebotomus perniciosus* (*n* = 594) | | | *Phlebotomus sergenti* (*n* = 4) | | | *Phlebotomus perfiliewi* (*n* = 3) | | | *Phlebotomus neglectus* (*n* = 1) | | |
| | Unfed | Blood fed | Gravid | Unfed | Blood fed | Gravid | Unfed | Blood fed | Gravid | Unfed | Blood fed | Gravid | Unfed | Blood fed | Gravid |
| Site A (2017) | 60 | 1 | 1 | 153 | 4 | | | | | | | | 1 | | |
| Site B (2018) | 121 | | 33 | 43 | 6 | 5 | | | | | | | | | |
| Site C (2018) | 137 | 11 | 5 | 1 | | | 1 | | | | | | | | |
| Site D (2018) | 239 | 4 | 5 | 7 | | 2 | 2 | 1 | | | | | | | |
| Site E (2018) | 587 | 21 | 39 | 223 | 128 | 22 | | | | 2 | | 1 | | | |
| Total | 1,144 | 37 | 83 | 427 | 138 | 29 | 3 | 1 | | 2 | | 1 | 1 | | |

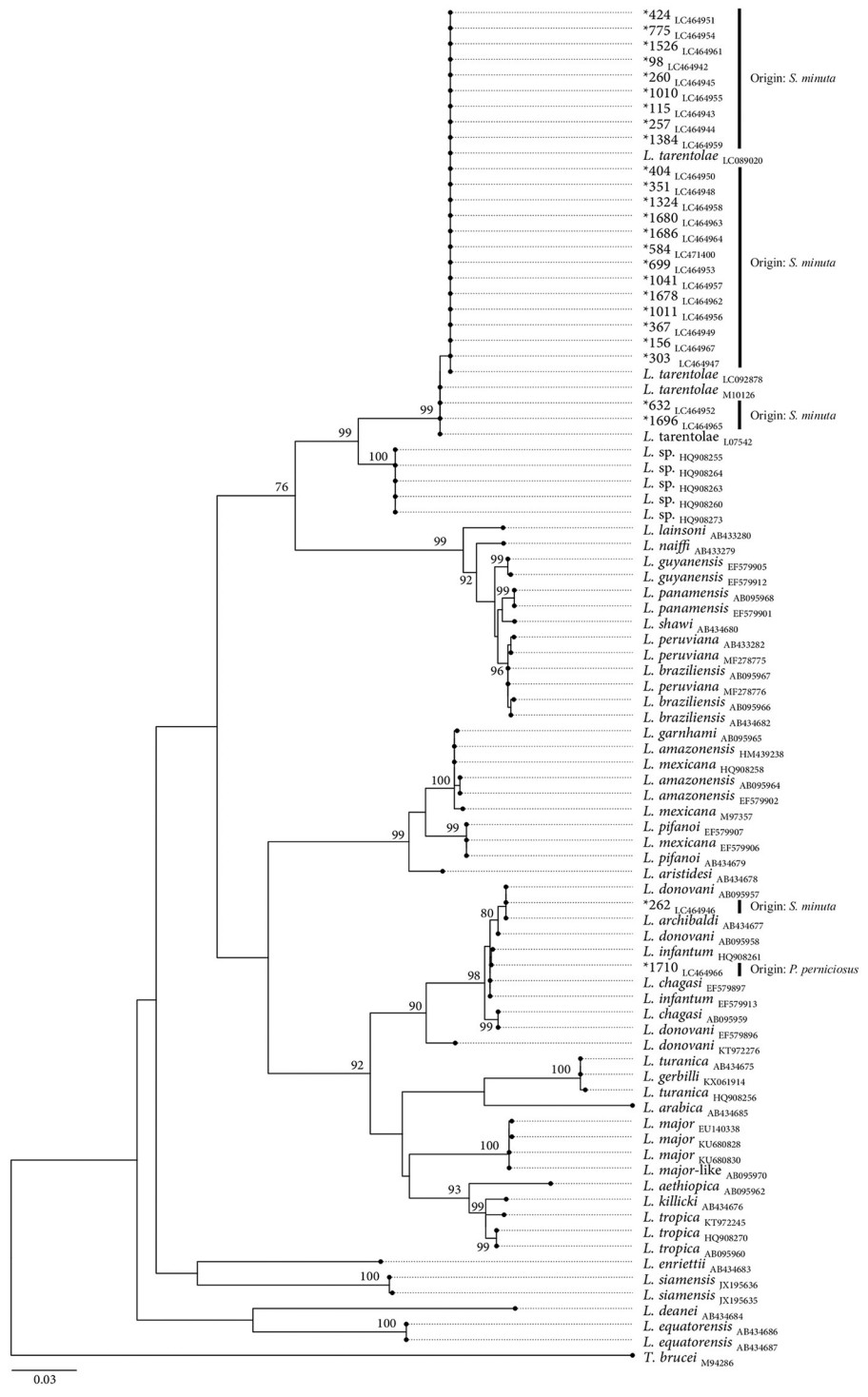

**Fig 2. Maximum likelihood phylogenetic tree based on unambiguous multiple-*Leishmania* cytB sequences alignment, using the GTR+G+I model of evolution.**

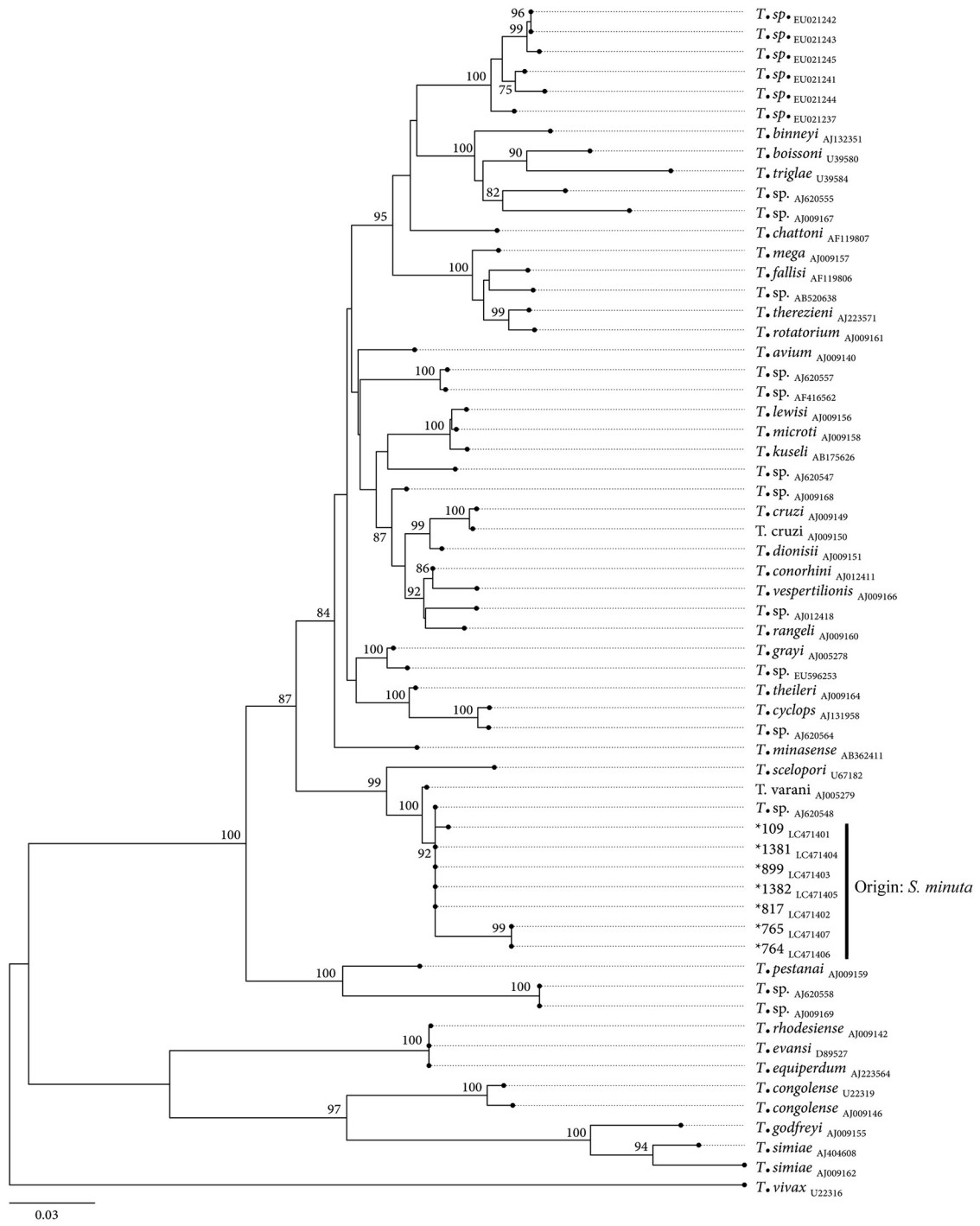

**Fig 3. Maximum likelihood phylogenetic tree based on unambiguous multiple-*Trypanosoma* SSU-rDNA sequences alignment, using the GTR+G+I model of evolution.**

At specific branch nodes bootstrap values (from 1000 random replicates of the original dataset) ≥ 75% are shown. The size bar indicates the number of nucleotide substitutions per site. The tree was rooted using as an outgroup a reference sequence of *Trypanosoma brucei brucei* (M94286). The sequences obtained in this study are identified with "*" and their respective accession numbers underscored.

At specific branch nodes bootstrap values (from 1000 random replicates of the original dataset) ≥ 75% are shown. The size bar indicates the number of nucleotide substitutions per site. The tree was rooted using as an outgroup a reference sequence of *Trypanosoma vivax* (U22316). The sequences obtained in this study are identified with "*" and their respective accession numbers underscored.

## Identification of sand fly blood meals

The majority of blood-fed females were captured in site E (149/176; *n* = 128 *P. perniciosus* and *n* = 21 *S. minuta*) and underwent statistical analysis, whereas the remainder of 27 blood-fed specimens caught at the other sites (A, *n* = 4 *P. perniciosus*, *n* = 1 *S. minuta*; B, *n* = 6 *P. perniciosus*; C, *n* = 11 *S. minuta*; D, *n* = 4 *S. minuta*, *n* = 1 *P. sergenti*) were not statistically analysed (Table 2). Host mitochondrial *cytB* was successfully amplified from 169 out of 176 engorged females (96%), and the vertebrate hosts of 108 blood meals (64%) were distinctly identified (S2 Table).

Wild rabbit (*Oryctolagus cuniculus*) (*n* = 28) and human (*Homo sapiens*) (*n* = 24) were the most frequent blood sources. Other vertebrate hosts also found were, *i.e.*, goat (*n* = 16), horse (*n* = 13) pig (*n* = 9), dog (*n* = 9), chicken (*n* = 3), cow (*n* = 3), cat (*n* = 1), donkey (*n* = 1) and rat (*n* = 1). Blood sources for each sand fly species and site of capture are shown in Fig 4.

Identification of blood source was successfully achieved in two out of the four engorged *S. minuta* specimens positive for *L. tarentolae* DNA, as being human (*Homo sapiens*) and donkey (*Equus asinus*); whereas the host mitochondrial *cytB* partial gene could not be amplified from the blood-fed *P. perniciosus* positive to *L. donovani* complex DNA.

Wild rabbits (27/82; 32.9%) represented the most preferred mammal species for *P. perniciosus*, while *S. minuta* mainly fed on humans (16/25; 64%). Results of the statistical analysis on frequency of blood sources for *P. perniciosus* and *S. minuta* from site E are summarized in S3 Table, where a statistically significant higher frequency of blood meal on rabbit by *P. perniciosus* compared to *S. minuta* ($\chi^2$ = 183.1; *P* = 0.01), and on human by *S. minuta* compared to *P. perniciosus* ($\chi^2$ = 27.7; *P* < 0.001), was detected.

## Discussion

Our study aimed to assess the presence of natural *Leishmania* infection in wild-caught sand fly species common in Sicily, Southern Italy; additionally, the source of blood meals of fed females was also determined to gain evidence on sand fly feeding habits and to allow the identification of potential/alternative reservoir hosts for *Leishmania*.

Four out of five sand fly species analysed in this study are proven vectors of human leishmaniosis. Among the genus *Phlebotomus*, *P. perniciosus* was the most abundant species, which is in agreement with other entomological surveys conducted in Southern Italy [13,29,37–38], and the presence of *L. donovani* complex DNA in *P. perniciosus* herein reported confirms the role of this species in the maintenance and spread of leishmaniosis in the Mediterranean area. Only a single *P. perniciosus* (0.2%; *n* = 594) tested positive for *L. donovani* complex. To the best of our knowledge, the sole data on the prevalence infection for *L. infantum* in *P. perniciosus* in Sicily came from a study conducted in the city of Catania where a higher infection rate (*i.e.*, 11%; 8/72) has been detected [13]. This difference may be related to the high endemicity of leishmaniosis in the area of Catania [11,13] and suggest that the risk of transmission of *Leishmania* can be high even in urban areas.

*Phlebotomus neglectus* and *P. perfiliewi* specimens were occasionally collected in suburban and rural environments, and *P. sergenti*, the vector able to transmit *L. tropica*, was also sporadically collected, with its presence limited to the eastern side of the Island [12–14].

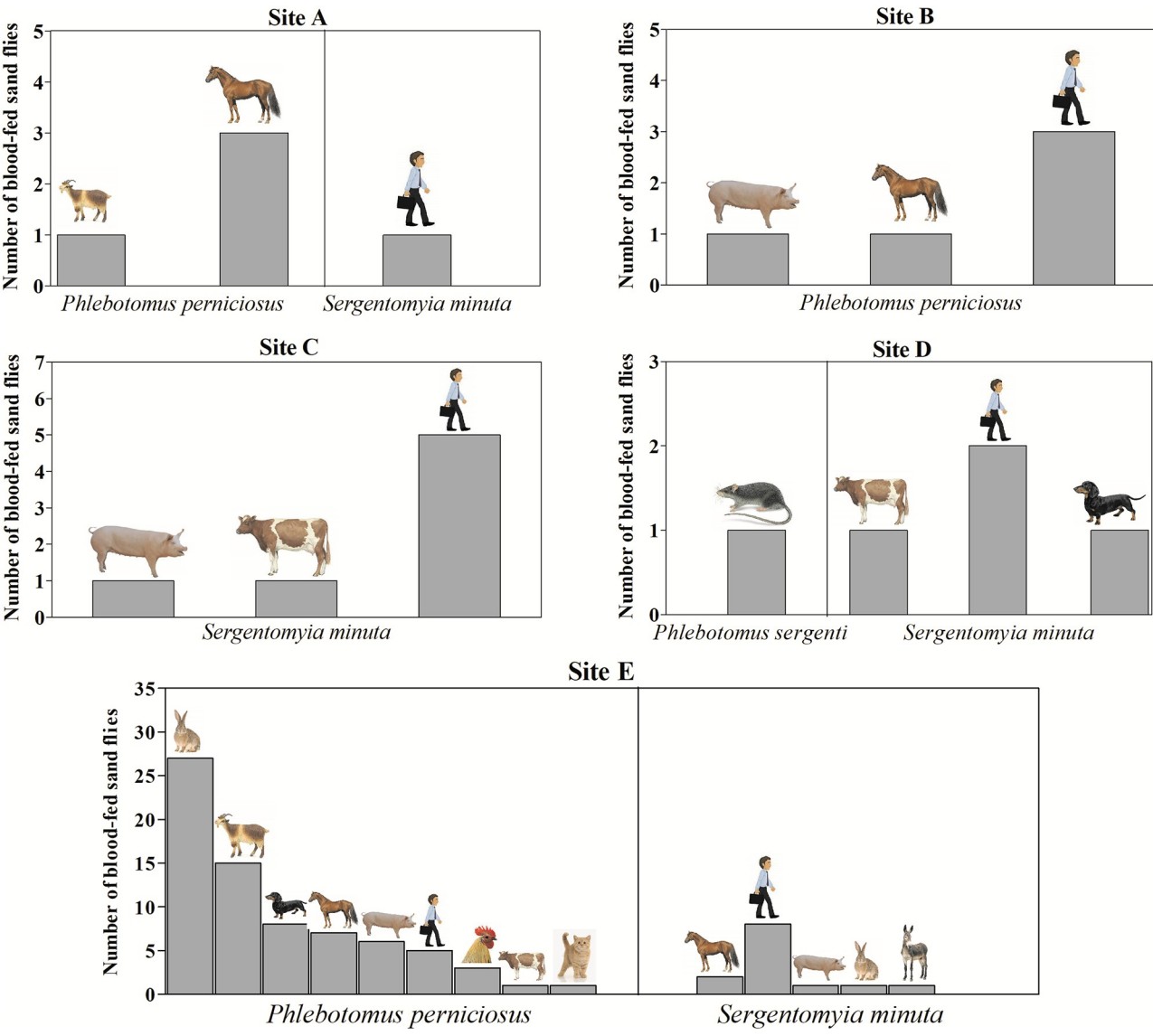

**Fig 4. Vertebrate hosts identified per each sand fly species in selected sampling sites (A-E).**

The detection of *L. donovani* complex DNA in a *S. minuta* unfed female herein recorded, spurs to better investigate the potential role of this species in the circulation and eventually transmission of the parasite [39]. The positivity of this sand fly species may be correlated to its feeding behaviour and the large availability of mammalian reservoir hosts positive to *Leishmania* in endemic foci [38]. Although *Sergentomyia* species mainly feed on cold-blooded vertebrates, its sporadic/opportunistic anthropophilic feeding behaviour has been already suggested [4] and, accordingly, a higher human blood preference has been herein statistically demonstrated. Scientific evidence on the competence of *S. minuta* in transmitting *Leishmania* spp. to warm-blooded mammals is not available so far; however, it has recently been demonstrated that *L. donovani*, *L. infantum* and *L. major* are not able to develop late-stage infections in *Sergentomyia schwetzi* sand fly species [40].*Sergentomyia* spp. are widespread Mediterranean sand fly species, and proven vectors of reptile *Leishmania* species (*e.g.*, *L. tarentolae*) non-pathogenic to humans. The positivity of

*S. minuta* for *L. tarentolae* herein recorded (2.1%; 26/1264) confirms that this species feeds on cold-blooded animals, as this protozoan parasite is widespread in Gekkonidae species [41].

In Italy, the presence of *Trypanosoma* spp. in sand flies has been little reported. In particular, *Trypanosoma platydactyli* was described in *S. minuta* [42], and the infection by a *Trypanosoma* belonging to *Trypanosoma theileri* group with very high homology to other trypanosomes detected in European cervids was recently reported in *Phlebotomus perfiliewi* [43]. In the Mediterranean area, *Trypanosoma nabiasi*, a rabbit trypanosome, and its co-infection with *L. infantum* was found in *P. perniciosus* female sand flies caught in the context of human leishmaniosis outbreak in Madrid [44], and natural infection of sand flies by trypanosomes of lizards, amphibians, birds and rodents has been already reported mainly from the American continent and Asia [45–48]. Despite *Trypanosoma* spp. being protozoan parasites transmitted by hematophagous insects [49–50] the potential of phlebotomine sand flies in transmitting trypanosomes is still unclear. The findings herein obtained, suggest that *S. minuta* might potentially be a vector of both *Leishmania* and *Trypanosoma* parasites in southern Italy. Nevertheless, the detection of the trypanosomatid DNA is not sufficient to incriminate the species as a competent vector since it may simply originate from a blood meal without undergoing further development and/or transmission. The detection of *L. donovani* complex DNA and, in addition, the detection of reptilian *Trypanosoma* DNA within *S. minuta* indicates that more attention is required when identifying parasitic organisms by PCR within sand fly vectors in areas where leishmaniosis is endemic. As a matter of fact, it is important to highlight that, when *Trypanosoma* and *Leishmania* species are present in the same geographical area, mixed infections could appear within the same host and/or vectors representing a challenge to their diagnosis [51–53].

In regard to the detection of *L. donovani* complex sequences in both *P. perniciosus* and *S. minuta*, a species-specific identification cannot be concluded based on *cytB*. In fact, the monophyly of *L. infantum* and *L. donovani* species has not been consistently shown [29].

Concerning the identification of blood meal sources in this study, several vertebrate hosts have been recognized as blood sources in the most abundant sand fly species (*P. perniciosus*, *S. minuta*), confirming the opportunistic feeding behaviour of these hematophagous insects [4,18–19].

Knowledge of the host preferences of sand flies under natural conditions is essential to understand how host choice and blood-feeding behaviour of sand flies influence their vector capacity in leishmaniosis foci. Even though *Canis lupus familiaris* is regarded as the main domestic reservoir host for zoonotic leishmaniosis, it seems not to be the preferred species on which they feed [4,18]. The results from the current study show that despite three out of the five selected sites being shelters with high an availability of dogs (*i.e.*, from 300 to 500 dogs per site), canine DNA was not detected in engorged *P. perniciosus* specimens caught in any of them. This finding could however be biased by the broad use of insecticides and repellents on dogs as preventive measures against sand fly bites and *Leishmania* transmission. In this study, during the sand fly sampling period, about a third of all dogs hosted in the three shelters were treated with actives repellent against sand flies.

Of the total engorged females investigated in this study 85% were caught on a farm located in a rural area far away from human settlements. There a large variety of vertebrate hosts were available, with dogs and rabbits (either domestic or sylvatic) being the most abundant species. At this site, wild rabbits represented the most preferred mammalian species for *P. perniciosus*, which agrees with what has already been reported in Spain [20–22]. In fact, high levels of anti-*P. perniciosus* saliva antibodies in wild rabbits suggested their exposure and attractiveness to sand flies [54]. The large availability of lagomorphs could contribute to the maintenance of high density of *P. perniciosus* in areas where these mammal species are abundant [54] and, the

role of lagomorphs in sustaining sylvatic *Leishmania* cycle independently from the domestic one has been strongly suggested [22,25].

The results obtained in the current study show a different blood feeding pattern of *S. minuta* compared to *P. perniciosus*, suggesting that the two sand fly species do not prefer the same vertebrate host. It is well-known that *S. minuta* feeds on cold-blooded reptiles; noteworthy, reptile DNA was not amplified in engorged females herein analysed, and only DNA of warm-blooded vertebrates was amplified. Interestingly, *S. minuta* caught in site E mainly fed on humans, though the presence of this vertebrate host inside the farm is limited to a few hours during the day, and no inhabitations were present nearby.

## Conclusions

The current survey describes *Leishmania* infection rate in *P. perniciosus* and *S. minuta* and documents, for the first time in Sicily, *Trypanosoma* DNA presence in *S. minuta* unfed females.

The findings herein reported, highlight the well-known role of *P. perniciosus* as competent vector of *L. infantum* in western Mediterranean area; also, the identification of blood meal sources suggests the importance of wild rabbits in the maintenance of *P. perniciosus* and their potential role as sylvatic reservoirs of *Leishmania*.

The role of wild animals in the epidemiology of leishmaniosis as well as that of *S. minuta* as a vector of *Leishmania* spp. to humans is worthy of future investigations to achieve efficient control strategies.

## Supporting information

**S1 Table. PCR protocols performed for detection and characterization of Trypanosomatidae DNA and blood source identification.** [a]Trypanossomatidae kinetoplast DNA. [b]Host mitochondrial DNA. [c]PCR product was previously diluted 1:50 in nuclease-free water. Abbreviations: *cytB*, cytochrome *b*; *ITS*-rDNA, ribosomal internal transcribed spacer DNA; *SSU*-rRNA, small subunit ribosomal DNA; bp, base pars.
(DOCX)

**S2 Table. CytB sequences analysis for each sample, percentages of BLAST identity and sequences accession numbers. A**: LC469805; LC469817; LC469821; LC469832; LC469834; LC469836; LC469839; LC469842-44; LC469846-48; LC469850-51; LC469853-56; LC469863; LC469865; LC469871; LC469875-76; LC469878; LC469882; LC469887. **B**: LC469829; LC469837; LC469840-41; LC469852; LC469857; LC469859-62; LC469867; LC469872; LC469874; LC469880-81; LC469891-93; LC469896; LC469898; LC469900-03. **C**: LC469807-10; LC469813-16; LC469818-20; LC469822-26. **D**: LC469806; LC469811; LC469827-28; LC469830; LC469833; LC469835; LC469869; LC469877; LC469884, LC469890; LC469897; LC469908. **E**: LC469838; LC469845; LC469849; LC469858; LC469868; LC469870; LC469873; LC469883; LC469899. **F**: LC469885-86; LC469888-89; LC469905; LC469907; LC469910-12. **G**: LC469812; LC469906; LC469909. **H**: LC469831; LC469894; LC469904. **I**: LC469866. **L**: LC469879. M: LC469895.
(DOCX)

**S3 Table. Differences in host preference within the most abundant sand fly species caught in site E.**
(DOCX)

## Author Contributions

**Conceptualization:** Emanuele Brianti.

**Data curation:** Jessica Maria Abbate.

**Formal analysis:** André Pereira, Gabriella Gaglio, Giulia Caracappa.

**Funding acquisition:** Gabriele Marino, Emanuele Brianti.

**Investigation:** Jessica Maria Abbate, Gabriella Gaglio, Giulia Caracappa, Emanuele Brianti.

**Methodology:** André Pereira, Francesca Arfuso, Gabriella Gaglio, Maria Rizzo, Giulia Caracappa.

**Project administration:** Matthias Pollmeier, Emanuele Brianti.

**Software:** André Pereira.

**Supervision:** Carla Maia, Salvatore Giannetto, Emanuele Brianti.

**Validation:** André Pereira.

**Visualization:** Gabriele Marino, Matthias Pollmeier.

**Writing – original draft:** Jessica Maria Abbate.

**Writing – review & editing:** Carla Maia, André Pereira, Francesca Arfuso, Gabriella Gaglio, Maria Rizzo, Giulia Caracappa, Gabriele Marino, Matthias Pollmeier, Salvatore Giannetto, Emanuele Brianti.

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
