## [Decision Letter · Decision Letter 0]

3 Jan 2020

PONE-D-19-33599

Leishmania  detection and blood feeding preferences of phlebotomine sand fly species common in the Mediterranean area

PLOS ONE

Dear Dr. Brianti,

Thank you for submitting your manuscript to PLOS ONE. After careful consideration, we feel that it has merit but does not fully meet PLOS ONE’s publication criteria as it currently stands. Therefore, we invite you to submit a revised version of the manuscript that addresses the points raised during the review process.

We would appreciate receiving your revised manuscript by Feb 17 2020 11:59PM. To enhance the reproducibility of your results, we recommend that if applicable you deposit your laboratory protocols in protocols.io, where a protocol can be assigned its own identifier (DOI) such that it can be cited independently in the future. For instructions see: http://journals.plos.org/plosone/s/submission-guidelines#loc-laboratory-protocols

We look forward to receiving your revised manuscript.

Kind regards,

Vyacheslav Yurchenko

Academic Editor

PLOS ONE

Additional Editor Comments:

I invite authors to revise the manuscript to address concerns of both reviewers.

2. We note that you are reporting an analysis of a microarray, next-generation sequencing, or deep sequencing data set. PLOS requires that authors comply with field-specific standards for preparation, recording, and deposition of data in repositories appropriate to their field. Please upload these data to a stable, public repository (such as ArrayExpress, Gene Expression Omnibus (GEO), DNA Data Bank of Japan (DDBJ), NCBI GenBank, NCBI Sequence Read Archive, or EMBL Nucleotide Sequence Database (ENA)). In your revised cover letter, please provide the relevant accession numbers that may be used to access these data. For a full list of recommended repositories, see http://journals.plos.org/plosone/s/data-availability#loc-omics or http://journals.plos.org/plosone/s/data-availability#loc-sequencing

3. In your Methods section, please provide additional location information of the collection sites, including geographic coordinates for the data set if available.

4. In your Methods section, please provide additional information regarding the permits you obtained for the work. Please ensure you have included the full name of the authority that approved the collection sites access and, if no permits were required, a brief statement explaining why.

5. Thank you for stating the following in the Financial Disclosure section:

'Sand flies collection in 2018 season has been partially funded by Bayer Animal

Health. The founder did not play any role in the study design, data collection and analysis, decision to publish and preparation of the manuscript.

Molecular analysis on sand flies were partially founded by the grant Research and Mobility no.015063 awarded by the University of Messina'

We note that one or more of the authors have an affiliation to the commercial funders of this research study: Bayer Animal Health GmbH

6. We note that Figure 1 in your submission contains map/satellite images which may be copyrighted.

Reviewers' comments:

Reviewer's Responses to Questions

**Comments to the Author**

1. Is the manuscript technically sound, and do the data support the conclusions?

Reviewer #1: Yes

Reviewer #2: No

2. Has the statistical analysis been performed appropriately and rigorously? 

Reviewer #1: Yes

Reviewer #2: Yes

3. Have the authors made all data underlying the findings in their manuscript fully available?

Reviewer #1: Yes

Reviewer #2: Yes

4. Is the manuscript presented in an intelligible fashion and written in standard English?

Reviewer #1: Yes

Reviewer #2: No

5. Review Comments to the Author

Reviewer #1: General comments:

Clearly written and comprehensible article bringing interesting information.

Title: “…common in the Mediterranean area” – the title is exaggerated – it is definitely not an analysis of sand flies common in the Mediterranean – only in the western part and still not all common species, rather only selected ones, which were common on localities (in Sicily) where the authors made their catches (over only two seasons). PLS change “…common in Sicily”. Introduction should, therefore, also focus primarily on Sicily and, where appropriate, on southern Italy. Most parts describing the faunistic situation should be move from Intro to Discussion.

It is really interesting that it was not possible to detect/identify any reptilian blood - it would like to comment on it with at least one sentence.

It would be appropriate to include a paragraph where the authors comment on the existence of different haplotypes. There were always two haplotypes for both Leishmania species (L.t. and L.d.), as well as for unnamed trypanosome species - at least they would like to mention or comment this situation. This is particularly interesting in the case of L.donovani / L.infantum - is it really possible to assume that both species were captured/detected?

Specific comments:

26. etc. n = XXX

35. etc. (27/82) – not clear, need explanation

36. blood rat

45. why only warm-blooded ?

68. etc. X% or X %

136/137: for more info about the used traps add “see Fig. 1”

160. individually ? = 1866 DNA extractions?

246. You should add the information that obtained sequences (of L.t. and L.d.) were not 100% identical – both species form two “haplotypes”

255. … the same for two left sequences (a monophyletic cluster with another L.t. strain)

259. add information about occurrence in: unfed / blood fed / gravid

264. add Tarentola annularis from Senegal

269. two (2)

270. T. varani / T. sp.

Tab 2. n (instead no); n = XY; It would be beneficial to add info about Leishmania/Trypanosoma infection into this table.

316 (express it also in percentage … 96 % - congratulation; and 64 %)

320/Tab 3. rooster ? / hen-chicken-fowl

Fig. 4. Unify column color/pattern for presented animals (use the same pattern for the same animal); y axis – better “No. blood fed females”

Tabs 3. and 4. Move to Supplementary material ???

453-4. It is/was not clear that Trypanosoma DNA was detected only in blood fed S.m. females??? This info must be add 327/8 and PLS add also info which blood sources were detected.

Reviewer #2: The manuscript describes detection and identification of trypanosomatids and bloodmeals in Sicilian sand flies. Despite the results are interesting, their interpretation is not correct and some parts of the manuscript (mainly Abstract and Introduction) need to be rewritten.

General or major comments

1. Some sand fly species studied are NOT „commonly present in the Mediterranean area“ or “common in the Mediterranean area”. Please change the info on lines 22, 125-126, 345-346. Title is also misleading, more specific one would be useful (e.g. “Identification of trypanosomatids and bloodmeals of sand flies in Sicily”).

2. It is necessary to mention (on lines 32-33) that Trypanosoma DNA found in S. minuta clusters with reptile Trypanosomatids. Therefore, finding of Trypanosoma DNA in S. minuta should not be interpreted as a support for S. minuta role in transmission of human pathogens. There is no single evidence that Trypanosoma DNA found originate from species infecting mammals. The sentence on lines 42-43 needs to be changed.

3. Surprisingly, authors did not find a single reptile blood in S. minuta which contradicts frequent finding of reptile trypanosomes. Again, this should be mentioned in the Abstract.

4. Introduction is too long. Text on lines 104-108 and 116-119 could be deleted as most of information is repeated again in Discussion.

Table 3 should include also the numbers of females with nonidentified bloodmeals: P. perniciosus (56?) and S. minuta (12?).

Table 4 should be moved into Supplement.

Minor comments

Single finding of Leishmania DNA in P. perniciosus should not be interpreted as confirmation of the vector role of this sand fly species (line 39) or highlighting its role in Mediterranean area (lines 455-456). It may agree with well-known role of this vector with circulation of L. infantum in western Mediterranean. Please, change sentences accordingly.

The second sentence of Introduction should be changed. It is not correct in two aspects: 1. There is a single bacterial pathogen transmitted by sand flies (not “plethora” as written by the authors) and number of sand fly-borne viruses is much lower than in mosquitoes. 2. Leishmania of subgenus Mundinia are very likely transmitted by biting midges, not by sand flies.

Lines 60-61: Text should be changed to clarify that both information is valid only for western Europe.

Line 65: clarify what do you mean by “in free territories”

Lines 73-80: Part about imported cases should be reduced or changed. The reference by Fotakis et al (2019) does not concern the Italy. Moreover, this study (published in a good journal) suffers of serious mistakes: only sand fly heads were used for Leishmania DNA and despite of this nonsense extremely high percentage of sand flies were claimed as positive! Such results cannot be trusted and should not be cited. If authors decide to mention/discuss the potential spread L. tropica in Italy, then there are two recent papers about this topic published in Int. J. Parasitol, showing high susceptibility of P. perniciosus and P. tobbi to L. tropica.

Lines 363-364: P. sergenti is NOT “the sole vector able to transmit L. tropica”. The vectorial role of P. arabicus is well described in many papers since 2003. Moreover, there are two papers on other sand flies as potential L. tropica vectors published this year, see above.

Lines 383-385: the information about potential of L. tarentolae to infect humans is misleading and should be deleted (together with both references).

Finding of reptile Trypanosoma in S. minuta is not unexpected (as said on line 404).

6. PLOS authors have the option to publish the peer review history of their article (what does this mean?). If published, this will include your full peer review and any attached files.

Reviewer #1: No

Reviewer #2: No

---

## [Author Response · Author response to Decision Letter 0]

4 Feb 2020

Dear Editor and Reviewers,

Thank you very much for reviewing our manuscript PONE-D-19-33599 entitled “Leishmania detection and blood feeding preferences of phlebotomine sand fly species common in the Mediterranean area”

We have addressed all your concerns and a detailed revision note to reviewers’ comments is provided below.

We hope that the revised manuscript is now suitable for publication in PLOSE ONE journal.

On the behalf of all Authors 

Yours sincerely

Emanuele Brianti

Editor’s comments

 Authors’ response: Done.

2. We note that you are reporting an analysis of a microarray, next-generation sequencing, or deep sequencing data set. PLOS requires that authors comply with field-specific standards for preparation, recording, and deposition of data in repositories appropriate to their field. Please upload these data to a stable, public repository (such as ArrayExpress, Gene Expression Omnibus (GEO), DNA Data Bank of Japan (DDBJ), NCBI GenBank, NCBI Sequence Read Archive, or EMBL Nucleotide Sequence Database (ENA)). In your revised cover letter, please provide the relevant accession numbers that may be used to access these data. For a full list of recommended repositories, see http://journals.plos.org/plosone/s/data-availability#loc-omics or http://journals.plos.org/plosone/s/data-availability#loc-sequencing

 Authors’ response: Done.

3. In your Methods section, please provide additional location information of the collection sites, including geographic coordinates for the data set if available.

Authors’ response: Done.

4. In your Methods section, please provide additional information regarding the permits you obtained for the work. Please ensure you have included the full name of the authority that approved the collection sites access and, if no permits were required, a brief statement explaining why.

Authors’ response: The study sites are not located in protected areas with constrains of flora and fauna protection, and no specific permission for sand fly collection is needed. Indeed, the suburban and/or rural areas from which sand fly specimens were collected were privately owned areas and, according to the Italian regulation, the owner’s consent to access and to sample in the area is enough. 

In material and methods section, We specified that the two different entomologic surveys were carried out in privately owned areas located in Sicily and, the access to collection sites as well as the sampling procedures were authorized by the owners.

5. Thank you for stating the following in the Financial Disclosure section:

'Sand fly collection in 2018 season has been partially funded by Bayer Animal

Health. The founder did not play any role in the study design, data collection and analysis, decision to publish and preparation of the manuscript.

Molecular analysis on sand flies were partially founded by the grant Research and Mobility no.015063 awarded by the University of Messina'

 We note that one or more of the authors have an affiliation to the commercial funders of this research study: Bayer Animal Health GmbH

 Authors’ response: Done.

 Authors’ response: Done.

 Authors’ response: Done.

Authors’ response: Done.

 6. We note that Figure 1 in your submission contains map/satellite images which may be copyrighted.

All PLOS content is published under the Creative Commons Attribution License (CC BY 4.0), which means that the manuscript, images, and Supporting Information files will be freely available online, and any third party is permitted to access, download, copy, distribute, and use these materials in any way, even commercially, with proper attribution. For these reasons, we cannot publish previously copyrighted maps or satellite images created using proprietary data, such as Google software (Google Maps, Street View, and Earth). For more information, see our copyright guidelines: http://journals.plos.org/plosone/s/licenses-and-copyright. We require you to either (a) present written permission from the copyright holder to publish these figures specifically under the CC BY 4.0 license, or (b) remove the figures from your submission:

 Please upload the completed Content Permission Form or other proof of granted permissions as an "Other" file with your submission. In the figure caption of the copyrighted figure, please include the following text: “Reprinted from [ref] under a CC BY license, with permission from [name of publisher], original copyright [original copyright year].”

Authors’ response: We changed the map in the Fig. 1 by replacing it with a figure obtained from a resource of public domain suggested by Editor.

Reviewers’ comments and Authors’ responses

Reviewer #1: General comments:

Clearly written and comprehensible article bringing interesting information.

Title: “…common in the Mediterranean area” – the title is exaggerated – it is definitely not an analysis of sand flies common in the Mediterranean – only in the western part and still not all common species, rather only selected ones, which were common on localities (in Sicily) where the authors made their catches (over only two seasons). PLS change “…common in Sicily”. Introduction should, therefore, also focus primarily on Sicily and, where appropriate, on southern Italy. Most parts describing the faunistic situation should be move from Intro to Discussion.

It is really interesting that it was not possible to detect/identify any reptilian blood - it would like to comment on it with at least one sentence.

Authors’ response: We thank the Reviewer for her/his positive comments. The title and the introduction were changed according to the suggestions. Briefly, the introduction has been amended by focusing mainly on southern Italy and Sicily, while some parts describing the faunistic situation in the area of the study were moved to discussion. 

It would be appropriate to include a paragraph where the authors comment on the existence of different haplotypes. There were always two haplotypes for both Leishmania species (L.t. and L.d.), as well as for unnamed trypanosome species - at least they would like to mention or comment this situation. This is particularly interesting in the case of L.donovani / L.infantum - is it really possible to assume that both species were captured/detected? 

Authors’ response: Regarding L. tarentolae cluster, and according to boosttrap values (based on 1000 resamplings of the original sequence data; Fig. 2) the existence of different haplotypes cannot be assumed. In addition, and despite different haplotypes within L. donovani complex cluster are suggested, the phylogenetic tree does not evidence the monophyly of L. donovani and L. infantum. In fact, the investigations on the phylogenetic relationships and the genetic diversity of L. donovani/L. infantum strains based in four gene markers have concluded that the segregation of L. donovani/L. infantum species-specific clusters is not consistent (Pereira et al. 2020, ref. 29). Therefore, based on this conclusion it is not possible to assume that both species were detected. 

Sentences:

In results: 

Lines 259-260: Despite the heterogeneity of the sequences, phylogenetic analysis has not evidenced an unambiguous existence of different haplotypes. 

Lines 262-265: Based on cytB phylogenetic tree, the sequences obtained from both specimens segregate independently. The existence of intra-groups was suggested within the L. donovani complex cluster, however none of them evidenced the monophyly of L. donovani/L.infanutm species.

In discussion: 

Lines 425-428 “In regard to the detection of L. donovani complex sequences in both P. perniciosus and S. minuta, a species-specific identification cannot be concluded based on cytB. In fact, the monophyly of L. infantum and L. donovani species has not been shown to be consistent (Pereira et al. 2020, ref 29) 

Specific comments:

26. etc. n = XXX 

Authors’ response: We checked and standardized it according to journal guidelines.

35. etc. (27/82) – not clear, need explanation

Authors’ response: The sentence has been rewritten and now reads “Twenty-seven out of 82 blood sources identified in fed females of P. perniciosus were represented by blood of wild rabbit,…”

36. blood rat 

Authors’ response: the sentence now reads “…the sole P. sergenti fed specimen took a blood meal on rat.”

45. why only warm-blooded ?

Authors’ response: We deleted “…warm-blooded…” in the sentence.

68. etc. X% or X % 

Authors’ response: We checked and standardized it as “X%”. 

136/137: for more info about the used traps add “see Fig. 1” 

Authors’ response: We added “Fig 1” in lines 128, 132.

160. individually? = 1866 DNA extractions? 

Authors’ response: Yes, DNA was individually extracted from each sand fly as specified in the Material and Methods section as following “Genomic DNA was extracted from portions (i.e., thorax and abdomen) of each sand fly specimen…”

246. You should add the information that obtained sequences (of L.t. and L.d.) were not 100% identical – both species form two “haplotypes”

Authors´ response: As previous answered, for L. tarentolae cluster, the boosttrap values of the phylogenetic tree does not support the existence of different haplotypes; similarly, and despite the two L. donovani complex sequences are different, the classification at species level cannot be assumed because the monophyly of these species was not consistently demonstrated.

255. … the same for two left sequences (a monophyletic cluster with another L.t. strain)

Authors´ response: Please see previous answer

259. add information about occurrence in: unfed / blood fed / gravid

Authors’ response: the required information has been added in the revised version.

264. add Tarentola annularis from Senegal

Authors’ response: Done.

269. two (2)

Authors’ response: Done

270. T. varani / T. sp.

Authors’ response: Sorry, but we are unable to catch this 

Tab 2. n (instead no); n = XY; It would be beneficial to add info about Leishmania/Trypanosoma infection into this table.

Authors’ response: We replaced “no” with “n=”. As regards to add information about Leishmania/Trypanosoma infection in the table, we believe that this may overload the table and unclear for the reader. However, we leave the final decision to the Editor, and if it is necessary, we will find the way to add also these data in the table. 

316 (express it also in percentage … 96 % - congratulation; and 64 %)

Authors’ response: Done.

320/Tab 3. rooster ? / hen-chicken-fowl

Authors’ response: changed to “chicken” 

Fig. 4. Unify column color/pattern for presented animals (use the same pattern for the same animal); y axis – better “No. blood fed females” 

Authors’ response: Fig. 4 has been amended as suggested by the Reviewer.

Tabs 3. and 4. Move to Supplementary material

Authors’ response: Agree, Tables 3 and 4 have been moved to Supplementary material.

453-4. It is/was not clear that Trypanosoma DNA was detected only in blood fed S.m. females??? This info must be add 327/8 and PLS add also info which blood sources were detected.

Authors’ response: Trypanosoma DNA was detected only in unfed Sergentomyia minuta females. We corrected the mistake in the text.

Reviewer #2: 

The manuscript describes detection and identification of trypanosomatids and bloodmeals in Sicilian sand flies. Despite the results are interesting, their interpretation is not correct and some parts of the manuscript (mainly Abstract and Introduction) need to be rewritten.

General or major comments

1. Some sand fly species studied are NOT „commonly present in the Mediterranean area“ or “common in the Mediterranean area”. Please change the info on lines 22, 125-126, 345-346. Title is also misleading, more specific one would be useful (e.g. “Identification of trypanosomatids and bloodmeals of sand flies in Sicily”).

Authors’ response: We agree with the Reviewer and the manuscript has been changed in line with her/his suggestions. “Mediterranean area” has been changed with “Sicily” and/or “Southern Italy” in lines 23, 120, 344, and the title amended as well.

2. It is necessary to mention (on lines 32-33) that Trypanosoma DNA found in S. minuta clusters with reptile Trypanosomatids. Therefore, finding of Trypanosoma DNA in S. minuta should not be interpreted as a support for S. minuta role in transmission of human pathogens. There is no single evidence that Trypanosoma DNA found originate from species infecting mammals. The sentence on lines 42-43 needs to be changed.

Authors’ response: We thank the Reviewer for her/his suggestions with which we agree. The sentences (Lines 32-33 and 42-43) have been changed accordingly.

3. Surprisingly, authors did not find a single reptile blood in S. minuta which contradicts frequent finding of reptile trypanosomes. Again, this should be mentioned in the Abstract. 

Authors’ response: The absence of reptile blood in blood-fed S. minuta specimens is now mentioned in the abstract.

4. Introduction is too long. Text on lines 104-108 and 116-119 could be deleted as most of information is repeated again in Discussion.

Authors’ response: We simplified the Introduction section according to Reviewer’s comments and suggestions.

Table 3 should include also the numbers of females with nonidentified bloodmeals: P. perniciosus (56?) and S. minuta (12?).

Authors’ response: We moved Table 3 to Supplementary material according to suggestion of Reviewer #1, and we added the numbers of females with non-identified blood meals (P. perniciosus n = 5; S. minuta n = 12) in the table. 

Table 4 should be moved into Supplement.

Authors’ response: Tables 3 and 4 were moved to Supplementary materials.

Minor comments

Single finding of Leishmania DNA in P. perniciosus should not be interpreted as confirmation of the vector role of this sand fly species (line 39) or highlighting its role in Mediterranean area (lines 455-456). It may agree with well-known role of this vector with circulation of L. infantum in western Mediterranean. Please, change sentences accordingly 

Authors’ response: We thank the Reviewer for her/his suggestion; the sentences were amended as suggested.

The second sentence of Introduction should be changed. It is not correct in two aspects: 1. There is a single bacterial pathogen transmitted by sand flies (not “plethora” as written by the authors) and number of sand fly-borne viruses is much lower than in mosquitoes. 2. Leishmania of subgenus Mundinia are very likely transmitted by biting midges, not by sand flies. 

Authors’ response: We thank the Reviewer for spotting these inaccuracies, the sentence has been changed accordingly. 

Lines 60-61: Text should be changed to clarify that both information is valid only for western Europe. 

Authors’ response: Done as suggested by the Reviewer.

Line 65: clarify what do you mean by “in free territories”. 

Authors’response: “Free territories” has been reworded in “non-endemic territories”.

Lines 73-80: Part about imported cases should be reduced or changed. The reference by Fotakis et al (2019) does not concern the Italy. Moreover, this study (published in a good journal) suffers of serious mistakes: only sand fly heads were used for Leishmania DNA and despite of this nonsense extremely high percentage of sand flies were claimed as positive! Such results cannot be trusted and should not be cited. If authors decide to mention/discuss the potential spread L. tropica in Italy, then there are two recent papers about this topic published in Int. J. Parasitol, showing high susceptibility of P. perniciosus and P. tobbi to L. tropica.

Authors’ response: We thank the Reviewer for her/his valuable comments; the reference Fotakis et al., 2019 has been deleted. Also, we read with interest and cited in the revised manuscript, the publication by Vaselek S and Volf P. 2019 showing high susceptibility of P. perniciosus and P. tobbi to L. tropica.

Lines 363-364: P. sergenti is NOT “the sole vector able to transmit L. tropica”. The vectorial role of P. arabicus is well described in many papers since 2003. Moreover, there are two papers on other sand flies as potential L. tropica vectors published this year, see above.

Authors’ response: We deleted “sole”. 

Lines 383-385: the information about potential of L. tarentolae to infect humans is misleading and should be deleted (together with both references).

Authors’ response: Agree, done. 

Finding of reptile Trypanosoma in S. minuta is not unexpected (as said on line 404). 

Authors’ response: We thank the Reviewer for notifying us of this inconsistency; “…unexpected…” has been deleted.

---

## [Decision Letter · Decision Letter 1]

10 Feb 2020

Identification of trypanosomatids and blood feeding preferences of phlebotomine sand fly species common in Sicily, Southern Italy

PONE-D-19-33599R1

Dear Dr. Brianti,

We are pleased to inform you that your manuscript has been judged scientifically suitable for publication and will be formally accepted for publication once it complies with all outstanding technical requirements.

With kind regards,

Vyacheslav Yurchenko

Academic Editor

PLOS ONE

Additional Editor Comments (optional):

Authors have adequately addressed all the reviewers' concerns and the manuscript can now be accepted for publication.

Reviewers' comments:

Reviewer's Responses to Questions

**Comments to the Author**

1. If the authors have adequately addressed your comments raised in a previous round of review and you feel that this manuscript is now acceptable for publication, you may indicate that here to bypass the “Comments to the Author” section, enter your conflict of interest statement in the “Confidential to Editor” section, and submit your "Accept" recommendation.

Reviewer #1: All comments have been addressed

2. Is the manuscript technically sound, and do the data support the conclusions?

Reviewer #1: Yes

3. Has the statistical analysis been performed appropriately and rigorously? 

Reviewer #1: N/A

4. Have the authors made all data underlying the findings in their manuscript fully available?

Reviewer #1: Yes

5. Is the manuscript presented in an intelligible fashion and written in standard English?

Reviewer #1: Yes

6. Review Comments to the Author

Reviewer #1: I am completely satisfied. The authors have adequately addressed all my comments and I feel that this manuscript is now acceptable for publication.

7. PLOS authors have the option to publish the peer review history of their article (what does this mean?). If published, this will include your full peer review and any attached files.

Reviewer #1: No

---

## [Editor Report · Acceptance letter]

25 Feb 2020

PONE-D-19-33599R1 

Identification of trypanosomatids and blood feeding preferences of phlebotomine sand fly species common in Sicily, Southern Italy 

Dear Dr. Brianti:

I am pleased to inform you that your manuscript has been deemed suitable for publication in PLOS ONE. Congratulations! Your manuscript is now with our production department. 

With kind regards,

on behalf of

Dr. Vyacheslav Yurchenko 

Academic Editor

PLOS ONE